# Polarization patterning in ferroelectric nematic liquids via flexoelectric coupling

Nerea Sebastián [1] ✉, Matija Lovšin [1,2], Brecht Berteloot[3], Natan Osterman[1,2], Andrej Petelin[1,2], Richard J. Mandle[4,5], Satoshi Aya [6,7], Mingjun Huang [6,7], Irena Drevenšek-Olenik [1,2], Kristiaan Neyts [3] & Alenka Mertelj [1]

The recently discovered ferroelectric nematic liquids incorporate to the functional combination of fluidity, processability and anisotropic optical properties of nematic liquids, an astonishing range of physical properties derived from the phase polarity. Among them, the remarkably large values of second order optical susceptibility encourage to exploit these new materials for non-linear photonic applications. Here we show that photopatterning of the alignment layer can be used to structure polarization patterns. To do so, we take advantage of the flexoelectric effect and design splay structures that geometrically define the polarization direction. We demonstrate the creation of periodic polarization structures and the possibility of guiding polarization by embedding splay structures in uniform backgrounds. The demonstrated capabilities of polarization patterning, open a promising new route for the design of ferroelectric nematic based photonic structures and their exploitation.

The recent experimental realization of ferroelectricity in a 3D fluid satisfies a century of speculation and is anticipated to be widely exploited in emerging technologies with significant societal impacts. Known for years and broadly exploited in modern display technologies, the standard nematic liquid crystal (NLC) state is uniaxial and non-polar. In 2017 two independent groups reported the occurrence of several N phases in two materials, RM734[1] and DIO[2]. The lower temperature phase of DIO was shown to exhibit ferroelectric order[2]. Subsequent works have identified the novel N phase in both compounds as a ferroelectric nematic ($N_F$) phase, in which inversion symmetry is broken, leading to large spontaneous electric polarization ($P_0 \sim 6\mu C/cm^2$ for RM734[3] and $P_0 \sim 5\mu C/cm^2$ for DIO[4]). It has been shown that the growth of polar order and splay deformation are connected in RM734, i.e., the N-$N_F$ transition is a ferroelectric-ferroelastic phase transition in which the growth of ferroelectric order is accompanied by the softening of the splay orientational elastic constant[5,6].

The technological relevance of NLCs relies on the multifaceted combination of optical anisotropy, responsiveness to electric fields, and standardized alignment control via the surface treatment of the confining media. Ferroelectric NLCS (FNLCs) have additional applicative potentialities owing to their unique combination of fluidity and spontaneous polarization[4,7,8], giant dielectric permittivity[7,9] and second-order non-linear optical properties[6,8,10–13] enabled by the lack of inversion symmetry. The exploitation of the latter is, at present, restricted by the limited ability to control and shape the polarization direction via surface boundary conditions. Besides orientational coupling, FNLCs are guided by polar coupling constraints[10,14,15], affected by surface charges around defects and highly restricted by the large depolarization field entailed by the large magnitude of the polarization vector **P**.

In analogy to piezoelectricity, in which strain induces polarization, splay and bend orientational field deformations in NLCs can cause electric polarization of the medium, although, in non-polar NLCs, such

[1]Jožef Stefan Institute, Ljubljana, Slovenia. [2]University of Ljubljana, Faculty of Mathematics and Physics, Ljubljana, Slovenia. [3]Liquid Crystals and Photonics Group, ELIS Department, Ghent University, Ghent, Belgium. [4]School of Physics and Astronomy, University of Leeds, Leeds, UK. [5]School of Chemistry, University of Leeds, Leeds, UK. [6]South China Advanced Institute for Soft Matter Science and Technology (AISMST), School of Emergent Soft Matter, South China University of Technology, Guangzhou, China. [7]Guangdong Provincial Key Laboratory of Functional and Intelligent Hybrid Materials and Devices, South China University of Technology, Guangzhou, China. ✉e-mail: nerea.sebastian@ijs.si

effects are small[16]. Here we experimentally demonstrate that in FNLCs, the flexoelectric coupling between deformation and polarization is strong and can be used to control the polarization direction. To demonstrate this, we design a series of polarization structures realized via periodic splay photoalignment-enabled patterns. We explore the differences between DIO and RM734 FNLC materials and investigate the stability of the observed structures via a simple model, including elastic and electrostatic torques.

## Results and discussion
### Monodomain structures and alignment quality
RM734 and DIO are filled at a temperature of the N phase into glass cells with patterned photoalignment and cell thickness (gap) $d$ (see "Methods"). Each cell contains an array of different photopatterned $1.3 \times 0.73$ mm$^2$ rectangles. Patterning via photoalignment prescribes at the cell surfaces anisotropic nonpolar in-plane alignment of the nematic director **n** (i.e., the direction of the average orientation of the molecules in the nematic phase) without an out-of-plane tilt. Two distinct electrode configurations were employed: (i) cells with top and bottom surfaces with uniform indium tin oxide (ITO) electrodes and (ii) cells with one uniform ITO surface and one surface patterned with interdigitated electrodes of 100 μm width and 1 mm gap. For both materials, the transition to the N$_F$ phase on cooling materializes in a two-step process, in which the phase transition is followed by a structural relaxation (**n** structure changes across the confining cell thickness often involving twisting[10,14]), which is conditioned by the surface anchoring. In the photopatterned cells, such structural relaxation occurs very close to the N–N$_F$ temperature for RM734, and the intermediate structure cannot systematically be stabilized. However, in the case of DIO, we have observed a temperature range of $\sim 5$–6 °C below the N$_S$–N$_F$ transition at which the undistorted state is stabilized. This structural relaxation occurs via the development of structured domain walls (cell background in Supplementary Fig. 4), and we will refer to the director structures as *before* and *after* structural relaxation.

We first evaluated orientational and polar surface anchoring of photoalignment in the N$_F$ phase for uniform patterns (Fig. 1). Remarkably, despite the lower anchoring strength expected for photoalignment, azimuthal surface anisotropy results in large monodomain patterns for both materials and those cells with uniform ITO at both surfaces (Fig. 1a–c, e). While the structural relaxation takes place in the surrounding unaligned areas, it skips the uniform patterns, which remain untwisted. Orientation is maintained for increasing cell gaps $d = 3$, 5, and 8 μm. We investigated the homogeneity of the polarization by Second Harmonic Generation microscopy (SHG-M) and interferometry microscopy (SHG-I). Interferometry conditions were set according to the main SHG signal contribution (Supplementary Fig. 5). The SHG interferogram (Fig. 1g) for RM734 shows for different areas that the phase of the SHG signal is the same, i.e., the polarization direction is preserved throughout the pattern.

Comparison with the pattern in Fig. 1d evidences the important role of the electrodes. In the case of charged surface impurities, floating conductive electrodes compensate the charges, effectively equalizing the potential across the cell and hindering polarization distortions. Conversely, in Fig. 1d, the absence of a uniform electrode in one of the surfaces leads to uncompensated charges and polarization distortions that lead to subdivision in numerous domains. On top and around the electrode, subdivision in domains is partially prevented. It should be noted that no appreciable differences were observed for both electrodes being floating, shorted, or grounded.

Occasionally defects formed at the transition are observed at the pattern edge before wall propagation. After the structural relaxation, they lead to domain walls surrounded by uniform alignment in RM734, but interestingly, in DIO to spike-like disclination lines (Fig. 2 shows the zoom-in of Fig. 1c framed area). Such lines unequivocally create a characteristic optical distortion around them that looks like a dragonfly, with long wings extending transversally to the defect and sharply ending at the defect-bending tip. We deduced the underlying directory structure by combining polarizing optical microscopy (POM) examinations (Fig. 2a–d) and diffractive transfer matrix method

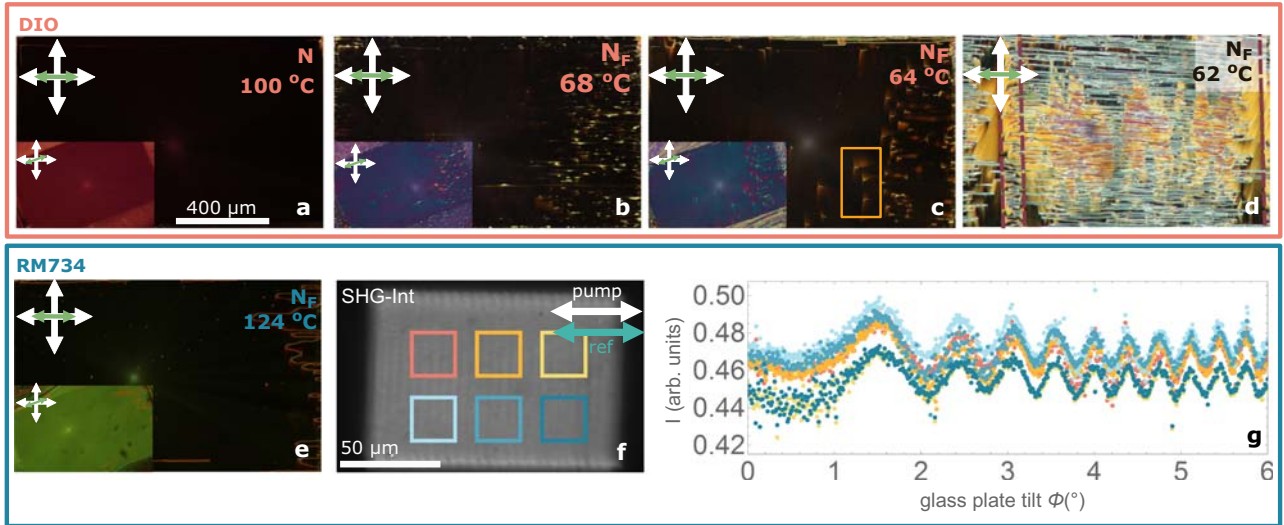

**Fig. 1 | Photopatterned millimeter range uniform polar domains.**
**a**–**c** $1.3 \times 0.7$ mm photopatterned uniform domains in DIO in the N phase (**a**), in the N$_F$ phase before final structural relaxation (**b**), and in the N$_F$ phase after structural relaxation in a 3 μm thick cell (**c**). The main images show the patterned aligning direction of the investigated area (horizontal as indicated by bidirectional green arrow) aligned with the crossed polarizers (white arrows), while the inset shows the same area under 20° anticlockwise rotation. Microphotographs show a uniform structure with some defects (see Fig. 2) arising from the unpatterned edges of the structured area. **d** Same photopatterned structure in a region of a cell where one of the surfaces has interdigitated electrodes shows that the absence of conductive

surfaces leads to an uncontrolled division in ferroelectric domains due to uncompensated charges and polarization distortions due to local impurities. Dashed maroon lines in d mark the position of the electrodes. In the broad region between them, only one of the confining cell surfaces has an ITO electrode. **e** Observations of the same photopatterned structure filled with RM734. **f** SHG-I image of the RM734 uniform structure and **g** SHG interferogram corresponding to the highlighted areas in (**f**). Measurements in the different areas show the same phase, indicating that the polarization direction is the same throughout the uniform pattern. Error bars smaller than the point size.

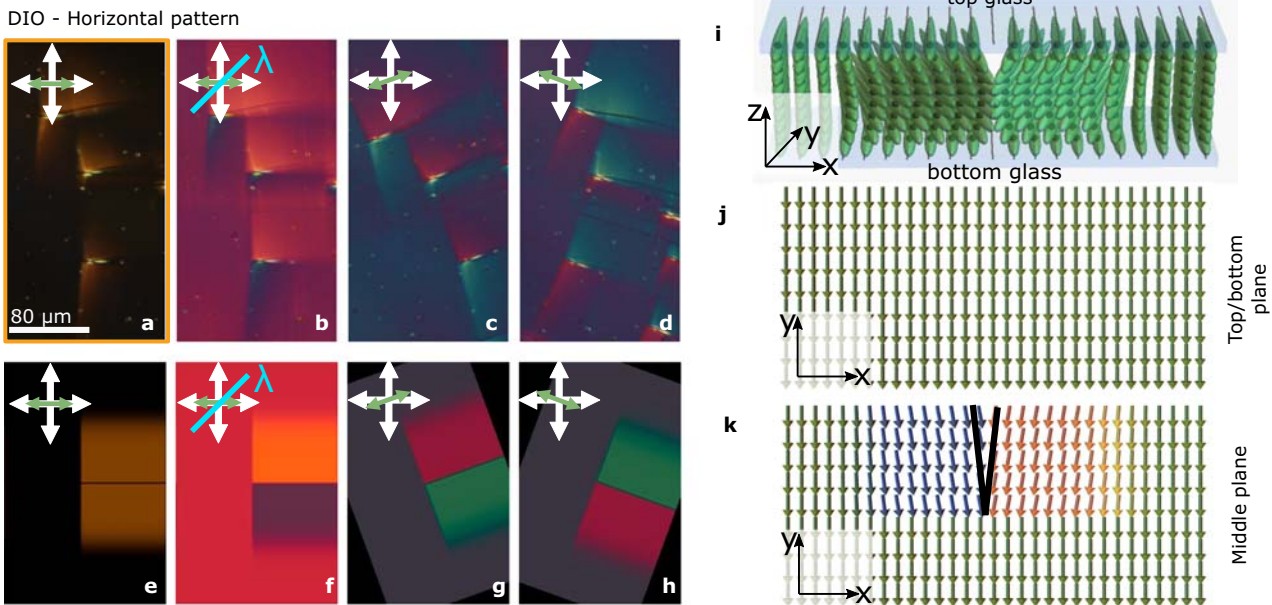

**Fig. 2 | Dragonfly-like splay deformation due to charged defect lines. a–d** Zoom-in POM images of the area highlighted in Fig. 1c at different geometries **a** extinction position, **b** with full wave plate at 45° and **c, d** with the sample rotated in opposite directions. Images show a distortion of the uniform structure around spike-shaped defect lines. Crossed bidirectional white arrows indicate the direction of the polarizer and analyzer, while the green arrow indicates the orientation of the photopatterned direction. **e–h** Dtmm simulations for the POM geometries in (**a–d**),

considering the structure described in (**i–k**). **i** 3D sketch of the director and polarization arrangement around a defect line in the uniformly patterned area. Here, light blue rectangles represent the cell surface glass plates. **j, k** The corresponding sketch in the cell surface plane (**j**) and in the middle of the cell plane (**k**), respectively. The structure twists across the cell thickness to accommodate the splay around the defect line. This splay director distortion reflects the electric charge of the defect line.

(dtmm[17]) optical simulations (Fig. 2e–h). At the surfaces, the director **n** is determined by the anchoring, and it twists towards the middle of the cell to accommodate an in-plane splay, as shown in Fig. 2i–k. Such splay distortion evidence that topological defect lines in the $N_F$, in this case, carry an electric charge.

**Control of polarization direction via flexoelectric coupling**

Considering the important role of splay deformations in the preceding nonpolar nematic phase[5,6] and stimulated by the quality of the observed alignment, we fabricated a series of periodic splay structures. The surface anchoring is shaped as $\mathbf{n}_s = (\sin(\vartheta_{surf}), \cos(\vartheta_{surf}))$, $\vartheta_{surf} = \vartheta_0 \sin(2\pi x/P)$, with $\vartheta_0$ ranging between 20° and 60° and the period $P = 2\pi/k$ ranging between 20 and 60 μm (Supplementary Fig. 6). The maximum splay curvature $k\vartheta_0$ of these patterns is then between 0.04 and 0.3 μm⁻¹. In the $N_F$ phase, the photoalignment results in large periodic arrays of lines with controlled alignment (Fig. 3a), and remarkably, this is so regardless of the ITO patterning conditions (Supplementary Fig. 7 and Video 1), which highlights the key role of splay $-\nabla \cdot \mathbf{P}$ "bound" charges in shaping the domain formation. We first focus on those patterns filled with DIO at a temperature of the $N_F$ phase before wall propagation. Careful inspection of POM textures during the $N_S$–$N_F$ transition shows, immediately after the transition, the formation of well-defined disclination lines running up-down the pattern where the splay changes sign (Fig. 3b). Combining POM observations together with dtmm simulations, we established the approximate $\mathbf{n}(\vec{r})$ structure (Fig. 3d–f. and Supplementary Fig. 11 to Fig. 13). While at both surfaces $\mathbf{n}(\vec{r})$ splays to follow the anchoring, towards the cell center it twists to recover a uniform structure (Fig. 3d–f). Such unsplay can be described by $\vartheta(z) = \vartheta_{surf} e^{(2z^2/d^2 - 1/2)/\xi}$ with $\xi = 0.2$.

SHG-M shows a rather uniform SHG signal within two consecutive disclination lines (Fig. 3c), in agreement with the described tendency towards uniform **n** orientation in the middle plane. The SHG signal (Fig. 3c) dependency on incoming laser polarization is in agreement with the observations in horizontal patterns and determines the

geometry for SHG-I measurements. Notably, disclination lines remain SHG inactive at any incoming polarization of the pump laser.

POM and SHG-M observations cannot distinguish between structures with the sign of **P** alternating with the splay direction or remaining the same. As the phase of the generated SHG light depends on the direction of **P**, regions with opposite polarity can be distinguished by changing the phase of a reference SHG signal. For DIO, the reference BBO crystal was placed so that the $SHG_{ref}$ polarization lies perpendicularly to the periodic stripes. SHG-I measurements (Fig. 4a, b) demonstrate that contiguous areas with opposite splay and separated by disclination lines have opposite directions of **P**, as the generated signal has a phase shift of 180° between them. These observations experimentally demonstrate how to exploit the flexoelectric coupling between polarization and orientational deformation for the design of polarization structures. Splay and bend director distortions in LCs can cause electric polarization of the medium, even in non-polar NLCs. This effect, coined flexoelectric effect by de Gennes[16] and originally described by R.B. Meyer[18], can be illustrated considering a nematic phase formed by wedge-shaped or crescent-shaped (bent-shaped) molecules with dipole moment along the long and short axis, respectively. In the first case, for a splay deformation, the minimization of the excluded volume results in the appearance of a macroscopic electric polarization proportional with $\mathbf{n}(\nabla \cdot \mathbf{n})$. Correspondingly, in the $N_F$ phase with spontaneous polarization **P**, the flexoelectric coupling is also allowed by the symmetry and the polar order couples with director deformations, implying that one direction of polarization is more favorable for a given splay deformation (Supplementary Fig. 10). Indeed, this effect spontaneously generates the periodic flipping of polarization in our studied periodic splay structures. Alternatively, when the system is described with only one order parameter (**P**), the effect is called flexodipolar coupling, i.e., coupling of **P** with deformations of **P**[19].

Splay deformation of **P** entails bound charges that can be calculated as $-\nabla \cdot \mathbf{P}$ (Fig. 4c) that generate a depolarization field, which

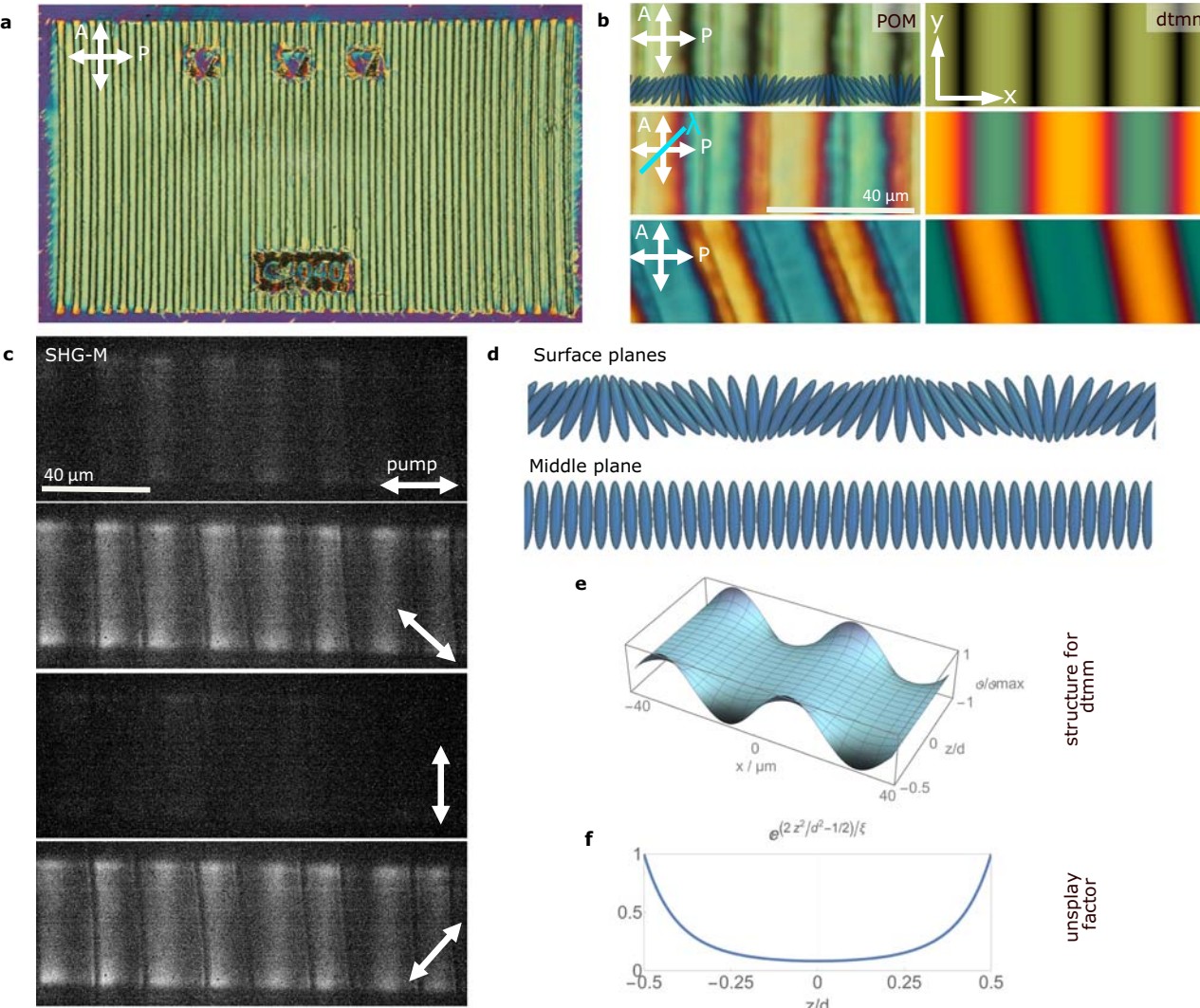

**Fig. 3 | Periodic splay patterns. a** Overview of the Splay-P40A40 periodic splay patterned area in a 3 μm cell filled with DIO in the $N_F$ phase, where the azimuthal photopatterned angle varies periodically as $\vartheta_{surf} = \vartheta_0 \sin(2\pi x/P)$ with maximum angle $\vartheta_0 = 40°$ and period 40 μm (splay curvature 0.1 μm⁻¹). The three top squares contain arrows pointing into lines in which **n** is vertical, and the bottom rectangle provides the structure label. **b** Zoom-in on the structure showing the POM observations (left) along with the corresponding dtmm simulations of the transmitted spectra (right) at different geometries (splay lines parallel to the analyzer, with a full lambda plate at 45° and sample rotated anticlockwise for 20°) taking into account the structure described through (**d**–**f**). **c** SHG-M images of a Splay-P40A40 pattern at different incoming laser polarizations, indicated by the bidirectional white arrow. **d** Schematic representation of the director orientation at the surface and in the middle plane of the cell used for dtmm simulations in (**b**). **e** Azimuthal angle $\vartheta(z,x)$ across 2 structure periods (x between −40 and 40 μm) and the confining cell thickness (z from −0.5 d to 0.5 d). **f** Profile for the reduction of the azimuthal angle across the cell thickness.

tends to suppress the splay, i.e., while splay is sustained in the surfaces due to the anchoring restrictions, the depolarization field suppresses it in the bulk as deduced via dtmm. This can be corroborated by means of a simple model to which we shall return in the next section.

Disclination lines (Fig. 3 and 4) separate regions with opposite **P** directions. We noticed that, although always appearing at the edge of the splay pattern, occasionally the disclination lines can move to the center of the splay region (i.e., they are not pinned to the surface), resulting in a slightly shifted structure (Supplementary Fig. 9). Taking into account that, due to the polarity of the phase, their topological charge must be integer and that, to avoid electrical charge in the core, twist deformations are the most favorable, we envision the possible structure of such lines as twist-like disclinations with topological charge ±1 as sketched in Supplementary Fig. 21. These structures are electrically charged, and thus, any splay structure asymmetry will cause their shift towards the center of the splay, which carries opposite electric charge (Fig. 4c).

On further cooling, the structural relaxation can also be detected in the photopatterned splay structure, characterized by the formation of domain walls, which initiate at the edges of the structure and propagate along the disclination lines. The optical transmission texture in DIO remains unaltered, with a slight distortion around the newly formed domain walls, in line with the above-described observations of the dragonfly-shaped distortions. In the case of RM734, disclination lines at the edge of the splay regions are also observed right at the transition (Supplementary Fig. 8). However, the subsequent structural relaxation takes place immediately after the N–$N_F$ transition, and although in the 3 μm cells alignment is reasonably preserved, in thicker (5 and 8 μm) cells a strong tendency to form π-twisted domains has been detected.

## Modeling of the periodic structure

To assess the stability of the periodic splay structures deduced from POM, we use a simplified model incorporating the depolarization field, whose effect in the $N_F$ case is relevant in defining the director

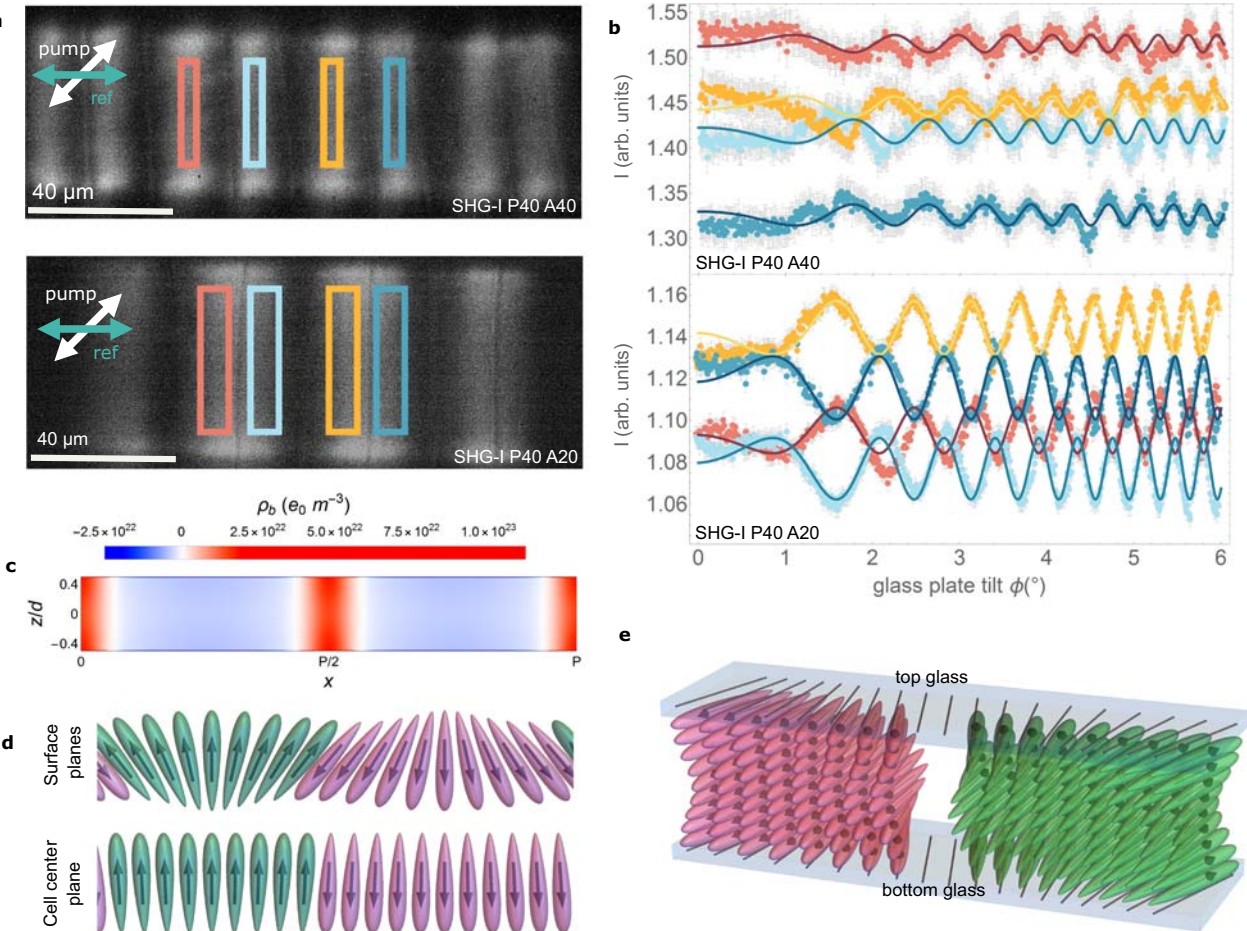

**Fig. 4 | Polarization patterning proved by SHG interferometry in periodic splay photopatterned structures. a** SHG interferometry images of the periodic-splay structures Splay-P40A40 and Splay-P40A20, with period 40 μm and maximum splay angle 40° and 20°, respectively (splay curvatures 0.1 and 0.05 μm⁻¹). The incoming polarization of the pump laser was at 45° as indicated by the white arrow, the reference signal was horizontal, and the signal was collected after a horizontal analyzer. **b** SHG interferograms corresponding to the highlighted areas in (**a**) with matching colors, where solid lines correspond to fits according to Supplementary Eq. 14 and error bars to computed SE. The SHG phase for two neighboring splay

domains is opposite, i.e., polarization lies in opposite directions for splay regions of opposite signs. **c** Calculated charge distribution $-\nabla \cdot \mathbf{P}$ across the cell thickness $d$, assuming the relaxed structure initiated from that described in Fig. 3 for two consecutive splay lines as depicted in (**d**). The relaxed structure is shown in Fig. 5a, b, and calculation details can be found in Supplementary Note VI. **d, e** Schematic representation of the surface and cell center planes (**d**) and 3D sketch (**e**) showing the flexoelectric coupling between the splay deformation and the polarization direction.

structure (Supplementary Note VI). First, we calculate local fields for the above-described structures, and subsequently, we employ the Euler–Lagrange formalism to compute the relaxed structure. For this, a simple model is considered, in which Frank–Oseen elastic torques[20] are counteracted by electric torques.

In general, FNLC can be described by two coupled order parameters, a nematic quadrupolar, i.e., tensor $\mathbf{Q} = S(\mathbf{n} \otimes \mathbf{n})$, and the electric polarization vector $\mathbf{P}$[21]. Here, $S$ is the scalar order parameter and $\mathbf{n}$ the director[16]. Here, we made the following assumptions: (i) $S$ is constant, so the nematic order can be described only by $\mathbf{n}$; (ii) $\mathbf{P} = \mathbf{P}_s + \varepsilon_0(\boldsymbol{\varepsilon} - \mathbf{I})\mathbf{E}$, where $\mathbf{P}_s = P_0 \mathbf{n}$; (iii) $P_0$ is constant; (iv) induced polarization anisotropy is neglected, and the dielectric tensor is thus taken as isotropic, $\boldsymbol{\varepsilon} = \varepsilon \mathbf{I}$; and (v) the orientation of the director at the surface is the same as prescribed by photopatterning. While the effective value of the dielectric constant measured by dielectric spectroscopy is large, i.e., of the order of 10000 (which mainly comes from the reorientation of $\mathbf{P}_s$)[21], the value of $\varepsilon$ as defined above only accounts for induced polarization and is expected to be of the order 100–1000.

By minimization of a Landau-de Gennes type free energy functional, we calculated stable structures for the splay patterns. The functional additionally includes the electrostatic potential[22] to account

for bound charges due to $-\nabla \cdot \mathbf{P}$ and free ions. Assuming that the dynamics of $\mathbf{n}$ (and $\mathbf{P}_s$) are much slower than that of free charges, then, during $\mathbf{n}$ relaxation, the local field $\mathbf{E}$ is given by the Poisson-Boltzmann equation $\nabla^2 \Phi_n = \beta^2 \sinh\Phi_n - \rho_{b,n}$ where $\Phi_n = e\Phi/k_B T$, being $\Phi(\boldsymbol{r})$ the electrostatic potential, and $\beta$ and $\rho_{b,n}$ account for the normalized free charges and volume charges respectively. Then the relaxation method can be used to minimize the part of the Landau-de Gennes free energy that depends on the director's orientation:

$$F_{\mathbf{n}} = \int \left( \tfrac{1}{2}K_1 |\mathbf{S} - \mathbf{S}_0|^2 + \tfrac{1}{2}K_2 Tw^2 + \tfrac{1}{2}K_3 |\mathbf{B}|^2 - \tfrac{1}{2}P_0 \mathbf{n} \cdot \mathbf{E} \right) dV$$

Here, $K_i$ ($i = 1,2,3$) are splay, twist, and bend elastic constants with corresponding deformations $\mathbf{S} = \mathbf{n}\nabla \cdot \mathbf{n}$, $Tw = \mathbf{n}\cdot(\boldsymbol{\nabla} \times \mathbf{n})$, $\mathbf{B} = \mathbf{n} \times (\boldsymbol{\nabla} \times \mathbf{n})$, and $\mathbf{E} = -\boldsymbol{\nabla}\Phi$. The flexoelectric coupling is added to the first term, where $\mathbf{S}_0 = \gamma \mathbf{P}/K_1$. The sign of the flexoelectric coefficient $\gamma$ determines which direction of $\mathbf{P}$ is favorable when a splay deformation $\mathbf{S}$ is present. The ideal splay curvature which would minimize the splay elastic energy is $\mathbf{n} \cdot \mathbf{S}_0$. The relaxation steps were performed with respect to $\vartheta(x,z)$ and $\varphi(x,z)$. At each step, $\mathbf{E}$ was recalculated using the linearized Poisson–Boltzmann equation (for $\Phi_n < 1, \nabla^2 \Phi_n = \beta^2 \Phi_n - \rho_{b,n}$).

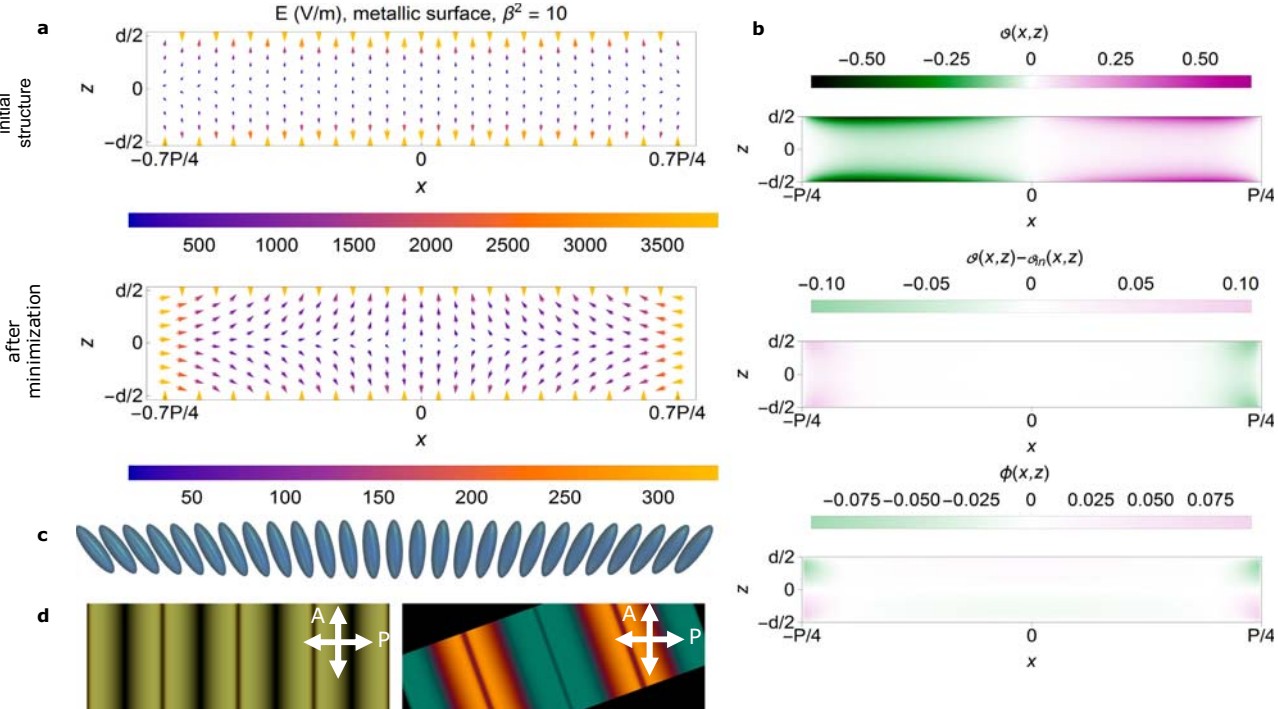

**Fig. 5 | Assessment of structure stability via electrostatic calculations.**
**a** Comparison of the local field for the initial structure (top) and the structure after relaxation (bottom) in the central region $(-0.7P/4 < x < 0.7P/4)$ as depicted in (**c**) for $\beta^2 = 10$ and $\varepsilon = 100$. The $x$ and $z$ coordinates are given in units of $\xi_b = 63$ nm. **b** Comparison of the structure before and after minimization in the central region for $\beta^2 = 10$ and $\varepsilon = 100$. The initial structure is given by the angles $\vartheta_{in}(x,z)$ and $\phi_{in}(x,z) = 0$ as described in Eq. 3 and Eq. 4 of the Supplementary Note VI. Angles in the plot are given in radians. **c** Sketch of the splay section corresponding to the local fields shown in (**a**). **d** Dtmm simulations corresponding to the relaxed structure defined by the angles $\vartheta(x,z)$ and $\phi(x,z)$ shown in (**b**) as observed between crossed polarizers (crossed white arrows) when oriented parallel to them or with the sample rotated 20°.

Mapping of the local fields for the initial and relaxed structures show large local fields and deformations at the edges of the splay pattern, where experimentally defects are observed (Fig. 5a). For a large enough ions density ($\beta^2 \geq 1$) the depolarization field originating from these regions is screened by the free ions, and thus has a negligible effect on the director structure in the center of the splay line. The reduction of the local depolarization field for the structure after minimization is significant (Fig. 5a. for $\beta^2 = 10$, $\varepsilon = 100$, i.e., a screening length of 20 nm). However, the director structure changes are small, being negligible in the azimuthal direction (Fig. 5b) and more notable in the appearance of an out-of-plane splay deformation that causes the redistribution of bound and free charges resulting in the reduction of the local fields. Dtmm calculations using the relaxed structure show minor differences with respect to those using the initial structure (Fig. 5d).

## Guiding polarization in uniform and bend environments

To assess the limiting splay curvatures needed to guide polarization, we first studied the inclusion of single splayed lines, with the same maximum splay angle (45 degrees) but differing width, into a uniformly aligned pattern (Fig. 6, $\vartheta_0 k = 0.1$, 0.05, and 0.025 $\mu m^{-1}$). Remarkable differences can be observed between both materials. In DIO, the splay curvatures comparable to those in periodic patterns result in equivalent polarization guiding, while lower splay curvatures (<0.05) are not sufficient for producing controlled polarization structures, as evidenced by the appearance of numerous disclination lines running along the splay photopatterned area (Fig. 6c–e). However, for RM734, all investigated splay strengths show the characteristic formation of disclination lines and subsequent domain wall formation at the edges of the splay structure (Fig. 6i–k). For both materials, the decrease of the splay curvature implies a decrease of the depolarization field, evidenced by the decrease of the unsplay of the structure

(Fig. 6f–h, l–n). SHG-M analysis in RM734 shows, in line with observations in the horizontal pattern, a maximum of SHG signal for incoming polarization of excitation laser along the main director orientation (Fig. 6o). Correspondingly, SHG-I demonstrates the alternation of polarization direction across the domain walls (Fig. 6p, q).

The observed differences between both materials should be attributed to distinct flexoelectric coupling properties and highlight the fact that for the future development of controlled alignment techniques, the characteristics of each employed $N_F$ material need to be taken into account in order to find the optimum splay configuration.

To test the possibilities of guiding polarization direction via splay photopatterning through a deformed background, we designed a circular domain of radius 500 $\mu m$ in which the director lies tangentially, having pure bend deformation in the inner and outer rings and an additionally overlaid set of circular splay lines ($\vartheta_0 = 45°$ and $P = 100$ $\mu m$) in the radial area between 150 and 350 $\mu m$ (Fig. 7a). Optical investigation and comparison with simulated dtmm transmission spectra show that director orientation accurately follows the prescribed orientation throughout the cell thickness in the pure bend areas, while the degree of in-plane splay is decreased towards the cell center similarly to what is observed in the periodic splay patterns (Fig. 7b–d and Supplementary Fig. 14). SHG inactive disclination lines are again evident, running tangentially along the splay lines (Fig. 7e) and separating regions of opposite polarization direction as demonstrated by SHG-I measurements (Fig. 7f, g).

## Conclusion and outlook

We have demonstrated how to exploit flexoelectric coupling between polarization and splay director deformations in FNLCs to create controlled polarization direction patterns of interest for multiple nonlinear photonic applications. Periodic splay patterns and single splay

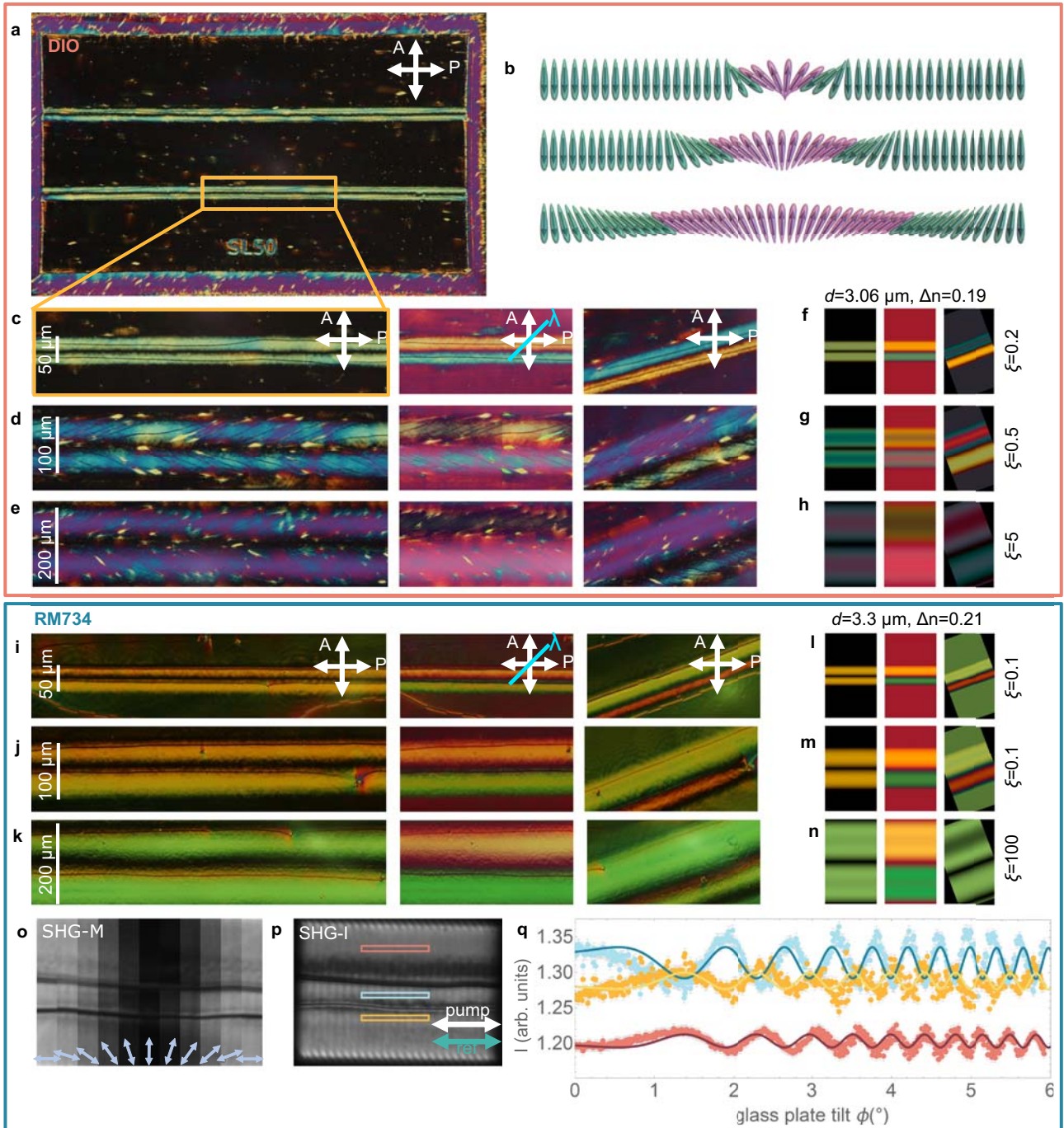

**Fig. 6 | Single splay lines embedded in the uniform background for guiding polarization. a** POM overview of the photopatterned $1.3 \times 0.7$ mm$^2$ structure filled with DIO. **b** Schematic representation of the three considered structures, i.e., uniform background in which single splay lines with $\vartheta_0 = 45°$ and $P = 50, 100,$ and $200$ μm are embedded ($\vartheta_0 k = 0.1, 0.05,$ and $0.025$ μm$^{-1}$). **c–n** Comparison of POM observations (**c–e, i–k**) and dtmm transmission spectra simulations (**f–h, l–n**) for the different single splay lines in DIO (**c–h**) and RM734 (**i–n**). For dtmm simulations in the splay regions, the azimuthal angle unsplays as $\vartheta = \vartheta_{surf} e^{(2z^2/d^2 - 1/2)/\xi}$, being $\vartheta = 0°$ along the splay lines, $\triangle n = 0.19$ for DIO, $\triangle n = 0.21$ for RM734 and $\xi$ for each case is shown in the figure. **c–e** Reduced splay curvature (**c** $0.05$ and **d** $0.025$ μm$^{-1}$) in DIO fail to produce controlled polarization domains and results in the appearance of multiple disclination lines oriented following the surface pattern. **f–h** dtmm

simulations show a reduction of unsplay with the reduction of the splay curvature. **i–k** For RM734, lower splay also results in well-oriented samples and controlled patterning of polarization directions. **l–n** dtmm simulations evidence the decrease of the unsplay structure, i.e., the reduction of the depolarization field. **o** shows a reconstruction of SHG-M images taken at different incoming pump laser polarizations. **p** SHG-I image of the same area depicted in (**o**). For SHG-I the incoming pump polarization and SHG reference signal were selected along the patterned lines, and the intensity was recorded with a horizontal analyzer. **q** SHG interferogram for the three rectangular areas highlighted in (**p**) showing opposite polarization direction in the area between both disclination lines (one splay direction) and outside them (opposite splay direction). Solid lines correspond to fits according to Supplementary Eq. 14 and error bars to computed SE.

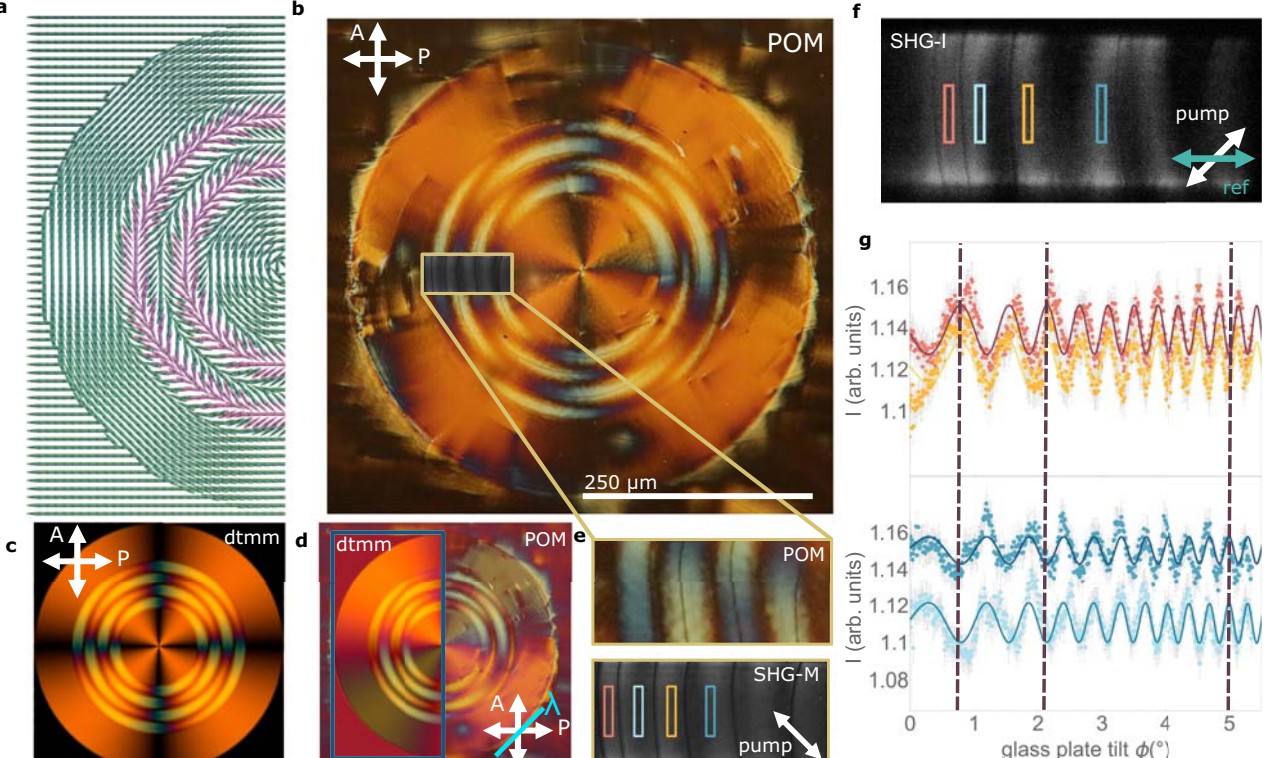

**Fig. 7 | Guiding of polarization with splay lines in a bend background.**
**a** Schematic of the photopatterned structure, in which two periods of a 4040 splay line are embedded tangentially into a pure bend circle deformation. Different arrowed colors highlight the alternation of regions with opposite polarization directions, as deduced in (**f**, **g**). **b** POM image of the splay in bend lines in a 3 μm cell filled with DIO. Crossed white arrows indicate the directions of the polarizers. The framed region shows the SHG-M image of that area. **c** Corresponding dtmm transmission spectra simulations considering a uniform bend structure in the outer and inner circles and a splay line deformation, with decreasing splay angle towards the center of the cell, equivalently constructed as the periodic splay lines considered in Fig. 3 showing a nice correspondence with experimental observations. **d** Comparison of POM and dtmm observations of the same structure with the addition of a full lambda plate at 45° with respect to the crossed polarizers, as indicated by the blue line. **e** Zoom image of the splayed region framed in (**b**) and corresponding SHG-M image with incoming laser polarization as marked by the white arrow. **f** SHG-I image of the same area as in (**e**). **g** SHG interferogram of the rectangular areas highlighted in (**f**) showing the opposite phase of SHG signal across each of the disclination lines, indicating the reversal of polarization direction for opposite splay lines. Solid lines correspond to fits according to Supplementary Eq. 14 and error bars to computed SE.

lines embedded in uniform or bent backgrounds all show alternating polarization directions between regions of opposite splay, with the appearance of SHG inactive disclination lines at the edge of the inscribed splay structure. The depolarization field created by bound charges due to $-\nabla \cdot \mathbf{P}$ causes the escape of the surface-prescribed splay structure towards a uniform orientation in the middle of the confining cell. The experimentally deduced structure is well explained by means of a simple model, which excludes the deformation around the charged defects. These results should serve as inspiration for further development of a model describing such defects in a general way, as done for the nonpolar nematic phase by Everts et al.[22]. Further investigations of the potentialities of photopatterning for FNLCs are expected to additionally enable stabilization of topological solitons, similarly as done for non-polar nematics[23]. The patterning of defects in N phases has also been recently exploited for reconfigurable steering of light[24]. The possibility of designing custom polarization structures, patterning SHG signal, and the prospect of steering and reconfiguring it by means of external electric fields is of great interest for applications in non-linear photonic devices well beyond the classical multi-billion technological implementations of NLC.

## Methods
### Materials
Synthesis of the liquid-crystalline material DIO (2,3′,4′,5′-tetrafluoro-[1,1′-biphenyl]−4-yl 2,6-difluoro-4-(5-propyl-1,3-dioxan-2-yl)benzoate) has been performed according to the description given in reference[12]. The

molecular structure, together with the phase sequence, is presented in Supplementary Fig. 1. Due to the 1,3-dioxane unit, the sample was always maintained below 120 °C to avoid changes in the molecular structure. On cooling from 120 °C, transition to Ns is observed at 83.9 °C and followed by the Ns–N$_F$ transition at 68.9 °C, as reported in reference[12]. Subsequent crystallization temperature was observed at temperatures lower than 60 °C, a temperature that differs depending on the cooling rate.

A description of the synthesis of the liquid-crystalline material RM734 can be found in ref. 1. The structure of RM734 (4-((4-nitro-phenoxy)carbonyl)phenyl-2,4-dimethoxybenzoate) and the phase sequence are presented in Supplementary Fig. 1. On heating, the crystalline phase melts directly into the nematic phase at 139.8 °C and transforms into the isotropic liquid phase at 187.9 °C. On cooling, the isotropic (I) to nematic (N) phase transition is followed by a nematic to ferroelectric nematic (N$_F$) transition at 132.7 °C, which crystallizes around 90 °C, a temperature that can vary depending on the cooling rate.

### Photopatterning
The LC director at the substrate surface was defined through patterned photoalignment. After ozone-plasma treatment, a mixture of 0.2 wt% Brilliant Yellow (Sigma-Aldrich) in dimethylformamide (DMF, Sigma-Aldrich) is used for spin coating (3000 rpm during 30 s). The substrates, with dimensions 1 in. by one inch, are either homogeneously coated with ITO or have an ITO electrode pattern. The substrates are then placed onto a hotplate for 5 min at 90 °C. With two coated

substrates, a cell is fabricated by placing glue and spherical glass beads near the edges of one substrate and placing the second substrate on top. When illuminated with linearly polarized light (UV or blue light), brilliant yellow provides the in-plane preferred orientation of nematic liquids perpendicular to the incident polarization. To achieve the desired patterns, the complete cell is then exposed to the illumination of a blue laser (Cobolt Twist, $\lambda$= 457 nm) that is modulated in polarization by an optical setup that contains a spatial light modulator (Holoeye Pluto 2). The pattern with 1920 by 1080 pixels is projected onto the cell, with the pixel voltage encoding the azimuthal angle of the linear polarization of the illumination. By moving the sample between subsequent illuminations, multiple patterns can be written in the same cell. For the patterns that provide a uniform alignment, an additional polarizer was inserted between the SLM and the LC cell. Detailed descriptions and schematics can be found in Supplementary Fig. 2

### Polarizing optical microscope and Berreman calculus

POM experiments were performed in a Nikon Eclipse microscope. Images and videos were recorded with a Canon EOS M200 camera. The sample was held in a heating stage (Instec HCS412W) together with a temperature controller (mK2000, Instec).

We used the "dtmm" open software package to calculate transmission spectra and color rendering. The package uses the Berreman $4 \times 4$ matrix method to compute the transmission and reflection spectra. The microscope's lamp spectrum was measured and used as an illuminant to calculate the transmission spectra. The latter is then converted to XYZ color space using CIE 1931 color-matching function. Then the linear RGB color from XYZ color space was computed, as described in the sRGB standard IEC 61966-2-1:1999. Finally, in order to obtain the final nonlinear RGB color values suitable for display or print, we applied the sRGB transfer function (gamma curve) using D65 white point. By this procedure, the input daylight light source spectra are converted to a neutral gray color in the case of uncrossed polarizers and no sample. To match the experimentally obtained images with the simulations, we performed in-camera white balance correction for daylight conditions to match the encoded D65 white point of the simulations allowing us to have a good agreement between the simulated and experimentally obtained images.

### SHG-M and interferometry

Interferometric SHG imaging is performed using a custom-built sample-scanning microscope (Supplementary Fig. 3). The laser source is an Erbium-doped fiber laser (C-Fiber A 780, MenloSystems) generating 785 nm, 95 fs pulses at a 100 MHz repetition rate. The average power was adjusted using an ND filter to 30 mW on the sample. The 1.2 mm beam diameter is expanded to 6.4 mm and focused on BBO crystal (Eksma Optics) to generate a reference SHG beam. In the case of SHG Microscopy, the BBO reference crystal is removed from the path. Polarization of the fundamental IR beam−marked with red−is perpendicular to the plane of the beam plane, whereas polarization of the reference SHG beam−marked with blue−is parallel to the plane. An off-axis parabolic mirror collimates the beams. Michelson interferometer is used for time compensation between the fundamental and the reference pulse. The phase of the reference pulse is adjusted finely with a glass plate mounted on a motorized rotator. A half-waveplate for 800 nm rotates the polarization of the fundamental IR beam in the plane. This polarization was chosen horizontally for the measurements of RM734 and with an angle of 45° for DIO, according to results described in Supplementary Fig. 5. A motorized dual-wavelength half-waveplate adjusts the incident polarization on the sample and, jointly with the analyzer in front of the camera, enables to perform polarization-resolved SHG. In the case of SHG-M images, the analyzer was removed to account for all contributions. A combination of galvo mirrors and a long-working distance objective (Nikon CFI T Plan SLWD, NA 0.3) is used to scan the focused beam in the sample plane. The scanning frequencies are much higher (a few 100 Hz) than the imaging frame rate (a few Hz).

A long-working distance 20× objective (Nikon CFI T Plan SLWD, NA 0.3) collects the light coming from the sample. A set of 700 nm short-pass and 400 nm band-pass filters eliminates the fundamental IR light and any possible fluorescence signal. The SHG-M and SHG-I images are finally acquired using a high-performance CMOS camera (Grasshopper 3, Teledyne Flir) with a typical integration time of 250 ms, dimensions $1920 \times 1200$ pixels, 24-bit depth, and 0.285 μm/pixel. SHG-I interferograms are obtained by computing the mean intensity and the standard error of the mean intensity of the areas of interest.

## Data availability

The authors declare that the data supporting the findings of this study are available within the text of this manuscript, including the Methods and the Supplementary information. Source data are available from the corresponding author upon request.

## Code availability

Code for diffractive transfer matrix method (dtmm) optical simulations is deposited in Github/Zenodo at "Andrej Petelin. (2020). IJS-ComplexMatter/dtmm: Version 0.6.1 (V0.6.1)", under accession code https://doi.org/10.5281/zenodo.4266242.

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

## Acknowledgements
N.S, A.M, I.D-O., M.L, N.O, and A.P acknowledge the support of the Slovenian Research Agency (grant numbers P1-0192, N1-0195, and PR-11214). K.N and B.B would like to acknowledge the support of the Research Foundation—Flanders (FWO) through grant number G0C2121N. R.J.M. acknowledges funding from UKRI via a Future Leaders Fellowship, grant No. MR/W006391/1. S.A. and M.H. acknowledge the National Key Research and Development Program of China (No. 2022YFA1405000) and the Recruitment Program of Guangdong (No. 2016ZT06C322).

## Author contributions
N.S., A.M., and K.N. designed and coordinated the work. N.S. performed POM observations. N.S. and A.M. carried out dtmm simulations. A.P. developed dtmm open software. M.L. performed SHG-M and SHG-I experiments. N.O. developed SHG-M and SHG-I setups. I.D.-O. assisted in the design and interpretation of SHG-M and SHG-I experiments. A.M. carried out the electrostatic calculation. K.N. and B.B. fabricated and characterized the photopatterned cells. R.J.M. synthesized RM734. S.A. and M.H. provided DIO. N.S. and A.M. prepared the initial draft of the paper, and all the authors made contributions to the final version.

## Competing interests
The authors declare no competing interests.
