## [Peer Review File · Nature Communications]

REVIEWER COMMENTS

Reviewer #1 (Remarks to the Author):

I congratulate authors with this Nie and important work. As the inventors of the FNLCs, a new breed of polar LCs with uninhibited fluidity, authors now demonstrate how polarization of such materials can be patterned. This yields insights into the studied materials as well as sets foundations for technological uses. The manuscript is well written, illustrations are clear, and the conclusions are well supported. Towards the end of main text, authors mention how their findings may open perhaps also patterning spatial charge distributions correlated with patterns of defects; such patterns indeed were recently created in non polar nematics and used for reconfigurable steering of light (doi:10.1038/s41563-022-01414-y), but to this reviewer it is not clear if all types of defects can be stabilized in this system, so perhaps only some can be possible given what authors describe. How about topological solitons that Neits group was patterning in non polar systems - could they be imprinted into this polar system too?

These are minor optional remarks and suggestions for discussion points - I recommend publication after (optional) revision.

Reviewer #2 (Remarks to the Author):

Review of manuscript "Polarization patterning in ferroelectric nematic liquids" written by N. Sebastián, M. Lovšin, B. Berteloot, N. Osterman, A. Petelin, R. J. Mandle, S. Aya, M. Huang, I. Drevenšek-Olenik, K. Neyts, A. Mertelj

The authors demonstrated the possibility to align of ferroelectric nematic liquid crystal (FNLCs) materials on photoalignment layers and exploited flexoelectric coupling between polarization and splay director deformations. They demonstrated FNLC alignment in uniform domains and on patterns containing periodically splayed structures embedded in uniform or bent backgrounds. In the former, they showed good quality alignment domains; in the later, they observed formation of alternating polarization domains between regions with opposite splays, and disclination lines at the edge of the splay structure.

Materials that can have stable FNLCs phase were recently discovered, which make them of particular interest from both fundamental study as well as application. The current manuscript contribute greatly for this topic, and thus is of interest to the community. However, to be accepted for publication at Nature Communications an essential revision of the manuscript is advised.

The main comment is that the paper very often refers to the Supplementary information document, which contains 26 pages and 25 Figures. It makes the manuscript very hard to read. The authors should consider compressing the Supplementary information, leaving only the most relevant figures, and revising how often they refer to them in the main text.

Other comments are:

1. Manuscript title does not fully reflect the reported work. The authors should consider revising the title.
2. p.2, section "Monodomain structures and alignment quality", first paragraph: authors discuss the role of ITO electrodes in creating uniform aligned domain. Were the ITO electrodes grounded during the experiment? Can authors comment on the effect of ITO electrode grounding on FNLC alignment?
3. p.6, "Modelling of the periodic structures". The model assumption "(iv) dielectric tensor is isotropic" is questionable. The authors gave an explanation later that the induced polarization is significantly lower in comparison with the contribution from the strong electric polarization. It should be rephrased, for example, as "induced polarization anisotropy is neglected".
4. There are several minor typos in the manuscript. Some of them are:
 - p.4, first column: main text refers to Fig. 3(g), however Figure 3 does not have panel g.
 - p.5. Fig. 4 caption. Sentence: "The SHG phase for two neighbouring splay domains is opposite, i.e. polarization lies in opposite directions for splay regions of opposite sign." repeated twice.
 - p.8, first column. Reference to Fig.6 e-g should be ref. to Fig.6 f-h

Thus, proper manuscript proofreading is required.

In summary, the manuscript can be considered for publication after a major revision.

In my view the manuscript can be recommended for a publication subject the authors address the following points.

1. The authors say

assumptions: (i) γ is constant, so the nematic order can be described only by \mathbf{n} ; (ii) $\mathbf{P} = \mathbf{P}_s + \epsilon_0(\epsilon - \mathbf{I})\mathbf{E}$, where $\mathbf{P}_s = P_0\mathbf{n}$; (iii) P_0 is constant; (iv) dielectric tensor is isotropic, $\epsilon = \epsilon\mathbf{I}$; and (v) the orientation of the director at the surface is the

In the formula for \mathbf{P} there are terms associated with the electric field induced polarization and spontaneous polarization P_s , however the LC director deformation may also result in the polarization. Why this formula does not include the flexopolarization?

2.

\mathbf{n} relaxation, the local field \mathbf{E} is given by the Poisson-Boltzmann equation and the relaxation method can be used

I think that the reader would benefit if the authors explicitly write down the Poisson-Boltzmann equation here.

3. The free energy written below seems missing the terms describing the flexopolarization and induced polarization interaction with electric field. Let us suppose that the spontaneous polarization is zero ($P_0=0$), than from the equation below it is seen that there is no interaction with the electric field at all. While there should be the electric field interaction with the induced polarization and with the flexopolarization.

to minimize the relevant part of the Landau-de Gennes free energy:

$$F_n = \int \left(\frac{1}{2}K_1|\mathbf{S} - \mathbf{S}_0|^2 + \frac{1}{2}K_2Tw^2 + \frac{1}{2}K_3|\mathbf{B}|^2 - \frac{1}{2}P_0\mathbf{n} \cdot \mathbf{E} \right) dV$$

Here, K_i ($i = 1,2,3$) are splay, twist, and bend elastic constants with corresponding deformations $\mathbf{S} = \mathbf{n}\nabla \cdot \mathbf{n}$, $Tw = \mathbf{n} \cdot (\nabla \times \mathbf{n})$, $\mathbf{B} = \mathbf{n} \times (\nabla \times \mathbf{n})$, and $\mathbf{E} = -\nabla\phi$. The flexoelectric coupling is added to the first term, where $\mathbf{S}_0 = \gamma\mathbf{P}/K_1$. The sign of the flexoelectric coefficient γ determines which direction of \mathbf{P} is favourable when a splay deformation

4. It looks like that in the manuscript (see the equation for F_n shown above) and Supplementary Information (eqn. (11)) \mathbf{S}_0 is a vector, on the other hand the Supplementary Information says $S_0 = \gamma\mathbf{P} \cdot \mathbf{n}$, which means S_0 is a scalar.

$$F_n = \int \left(\frac{1}{2}K_1|\mathbf{S} - \mathbf{S}_0|^2 + \frac{1}{2}K_2Tw^2 + \frac{1}{2}K_3|\mathbf{B}|^2 - \frac{1}{2}P_0\mathbf{n} \cdot \mathbf{E} \right) dV. \quad (11)$$

The relaxation steps were performed with respect to $\vartheta(x, z)$ and $\varphi(x, z)$. At each step, \mathbf{E} was recalculated using Eq.7. Here, K_i ($i = 1,2,3$) splay, twist, and bend elastic constants with corresponding deformations $\mathbf{S} = \nabla \cdot \mathbf{n}$, $Tw = \mathbf{n} \cdot (\nabla \times \mathbf{n})$, $\mathbf{B} = \mathbf{n} \times (\nabla \times \mathbf{n})$, and $\mathbf{E} = -\nabla\phi$. The flexoelectric term is included in the first term, where $S_0 = \gamma\mathbf{P} \cdot \mathbf{n}/K_1$ is the ideal splay curvature, which would minimize the splay elastic energy. The sign of S_0 determines the

5. To calculate the electric field the authors assume that the LC is isotropic fluid, the LC dielectric tensor is isotropic and has only one component.

$e^{(2x^2-0.5L^2)/(0.05L^2)}$), with the last term being discussed below. To simplify calculations, it was assumed the dielectric tensor is isotropic, $\epsilon = \epsilon \mathbf{I}$.

In fact the LC possesses the dielectric anisotropy, ϵ is a tensor. The LC dielectric anisotropy can be rather high. This anisotropy may significantly affect the local electric field, both the magnitude and the spatial profile of $\mathbf{E}(\mathbf{r})$. Isn't the above assumption an oversimplification? How big is the expected error when the LC is assumed to be an isotropic medium?

6. To calculate the electric potential the authors linearize the Poisson-Boltzmann equation.

Here, $\Phi_n = e\Phi/k_B T$. If $\Phi_n < 1$, as an approximation, a linearized Poisson-Boltzmann equation

How do we know that $\Phi_n < 1$? An estimate would be useful here.

7. Minor comment. "... described by R.B. Meyer,¹⁸ can be illustrated considering a nematic phase formed by **pear-shaped or bent-shaped**." Originally Meyer (ref [18]) called them wedge and crescent shape, see the picture from his work [18].

FIG. 1. (a) and (c) Unstrained nematic structures containing polar molecules. (b) Splayed structure in which splay and polarization are coupled by the wedge shape of the molecule. (d) Bent structure in which the bending and polarization are coupled by the crescent shape of the molecule.

RESPONSE TO REVIEWER COMMENTS

First, we would like to thank all referees for taking their valuable time to carefully read our manuscript and giving their important comments for improving it. Below, we address their concerns and answer to their questions accordingly.

Reviewer #1 (Remarks to the Author):

I congratulate authors with this Nie and important work. As the inventors of the FNLCs, a new breed of polar LCs with uninhibited fluidity, authors now demonstrate how polarization of such materials can be patterned. This yields insights into the studied materials as well as sets foundations for technological uses. The manuscript is well written, illustrations are clear, and the conclusions are well supported. Towards the end of main text, authors mention how their findings may open perhaps also patterning spatial charge distributions correlated with patterns of defects; such patterns indeed were recently created in non polar nematics and used for reconfigurable steering of light (doi:10.1038/s41563-022-01414-y), but to this reviewer it is not clear if all types of defects can be stabilized in this system, so perhaps only some can be possible given what authors describe. Ho about topological solitons that Neits group was patterning in non polar systems - could they be imprinted into this polar system too?

These are minor optional remarks and suggestions for discussion points - I recommend publication after (optional) revision.

We agree with the referee that studies with ferroelectric nematic liquid crystals such as those mentioned (C. Meng et al. doi:10.1038/s41563-022-01414-y and I. Nys et al. doi: 10.3390/cryst10090840) are highly interesting and a definitely the way to continue this research. Thus, we included a mention in the conclusions/outlook section. Structures with $k=1$ and $\Psi_0 = 0$ and $\Psi_0 = \pi/2$ have already been tested and are focus of current research. However, we can advance that the structures that we have observed are more complicated than in the case of non-polar nematics and strongly depend on the flexoelectric properties of the employed material. The line defects described in this study, separating the regions of opposite splay are systematically stabilized, although also depending on the amplitude and periodicity of the employed structure. There is definitely a lot of prospect research in this area, and we believe photopatterning will become an essential technique for the investigation and exploitation of ferroelectric nematics.

New added text:

These results should serve as inspiration for further development of a model describing such defects in a general way, as done for the nonpolar nematic phase by Everts et al.²² Further investigations of the potentialities of photopatterning for FNLCs is expected to additionally enable stabilization of topological solitons, similarly as done for non-polar nematics²³. Patterning of defects in N phases has also been recently exploited for reconfigurable steering of light²⁴. The possibility to design custom polarization structures,

Reviewer #2 (Remarks to the Author):

Review of manuscript "Polarization patterning in ferroelectric nematic liquids" written by N. Sebastián, M. Lovšin, B. Berteloot, N. Osterman, A. Petelin, R. J. Mandle, S. Aya, M. Huang, I. Drevenšek-Olenik, K. Neyts, A. Mertelj

The authors demonstrated the possibility to align of ferroelectric nematic liquid crystal (FNLCs) materials on photoalignment layers and exploited flexoelectric coupling between polarization and splay director deformations. They demonstrated FNLC alignment in uniform domains and on patterns containing periodically splayed structures embedded in uniform or bent backgrounds. In the former, they showed good quality alignment domains; in the later, they observed formation of alternating polarization domains between regions with opposite splays, and disclination lines at the edge of the splay structure.

Materials that can have stable FNLCs phase were recently discovered, which make them of particular interest from both fundamental study as well as application. The current manuscript contribute greatly for this topic, and thus is of interest to the community. However, to be accepted for publication at Nature Communications an essential revision of the manuscript is advised.

The main comment is that the paper very often refers to the Supplementary information document, which contains 26 pages and 25 Figures. It makes the manuscript very hard to read. The authors should consider compressing the Supplementary information, leaving only the most relevant figures, and revising how often they refer to them in the main text.

We agree with the referee that references to Supplementary information in the main manuscript affects its readability. As such, we minimized the mentions in the main text. Regarding the Supplementary information, we have combined several images together, to avoid repeated information in an attempt to reduce the length. However, we have been conservative about reducing the amount of content. We believe that any claim or statement in the manuscript should be sustained by reporting the corresponding experimental evidence. In this way, although the included Supplementary information Notes and Figures do not constitute the core of the reported investigations, they provide the necessary justification to the statements contained in the main manuscript. The additional advantage of SI for the reported research is that POM images can also be included in large format so that the reader can appreciate also the details. Although, of course, this increases the page count.

Other comments are:

1. Manuscript title does not fully reflect the reported work. The authors should consider revising the title.

Selecting the right title for a manuscript is always challenging. Although authors are not sure of what reviewer misses in the title, we took, for guidance, the summarizing sentence: *"The authors demonstrated the possibility to align of ferroelectric nematic liquid crystal (FNLCs) materials on photoalignment layers and exploited flexoelectric coupling between polarization and splay director deformations."* With that in mind, we propose the following revised title:

Polarization patterning in ferroelectric nematic liquids via flexoelectric coupling.

2. p.2, section "Monodomain structures and alignment quality", first paragraph: authors discuss the role of ITO electrodes in creating uniform aligned domain. Were the ITO electrodes grounded during the experiment? Can authors comment on the effect of ITO electrode grounding on FNLC alignment?

We investigated three different conditions: floating, shorted and grounded. No appreciable differences were observed. We agree that this is an important point to be considered and as such, we included the following clarification in the main manuscript text:

On top and around the electrode, subdivision in domains is partially prevented. It should be noted that no appreciable differences were observed for both electrodes being floating, shorted or grounded.

3. p.6, “Modelling of the periodic structures”. The model assumption “(iv) dielectric tensor is isotropic” is questionable. The authors gave an explanation later that the induced polarization is significantly lower in comparison with the contribution from the strong electric polarization. It should be rephrased, for example, as “induced polarization anisotropy is neglected”.

The simplified model presented in this manuscript qualitatively describes the system, and is explored to assess the stability of the structures deduced from the microscopy investigations. Taking into account that the induced part is expected to be lower than P_s , together with the lack of reliable dielectric anisotropy data at this moment, we opted for the most sensible approach of simplifying the model and neglect the induced polarization anisotropy, as the reviewer mentions. According to reviewer’s suggestion the sentence has been rephrased:

i) S is constant, so the nematic order can be described only by \mathbf{n} ; (ii) $\mathbf{P} = \mathbf{P}_s + \epsilon_0(\epsilon - \mathbf{I})\mathbf{E}$, where $\mathbf{P}_s = P_0\mathbf{n}$; (iii) P_0 is constant; (iv) induced polarization anisotropy is neglected and dielectric tensor is thus taken as isotropic, $\epsilon = \epsilon\mathbf{I}$; and (v).....

Additionally we added the following clarification into the SI:

If the dielectric tensor is taken anisotropic, i.e. $\epsilon = \epsilon_{\perp}\mathbf{I} + \Delta\epsilon(\mathbf{n} \otimes \mathbf{n})$, there would be an additional term $-\frac{1}{2}\Delta\epsilon\epsilon_0(\mathbf{n} \cdot \mathbf{E})^2$, which is orders of magnitude smaller than the term $-\frac{1}{2}P_0\mathbf{n} \cdot \mathbf{E}$, due to the large polarization values ($\frac{P_0}{\epsilon_0} \sim 5 \cdot 10^9 \text{ V/m}$). So from this point of view, neglecting the dielectric anisotropy is justified.

4. There are several minor typos in the manuscript. Some of them are:

- p.4, first column: main text refers to Fig. 3(g), however Figure 3 does not have panel g.
- p.5. Fig. 4 caption. Sentence: “The SHG phase for two neighbouring splay domains is opposite, i.e. polarization lies in opposite directions for splay regions of opposite sign.” repeated twice.
- p.8, first column. Reference to Fig.6 e-g should be ref. to Fig.6 f-h

Thus, proper manuscript proofreading is required.

We thank the referee for spotting these mistakes. They have been corrected and a thorough proofreading was carried out.

In summary, the manuscript can be considered for publication after a major revision.

 Reviewer #3 (Remarks to the Author): see the attached file

We included comments from the attachment here:

In my view, the manuscript can be recommended for a publication subject the authors address the following points.

1. The authors say

Here, we made the following assumptions: (i) S is constant, so the nematic order can be described only by \mathbf{n} ; (ii) $\mathbf{P} = \mathbf{P}_s + \epsilon_0(\epsilon - \mathbf{I})\mathbf{E}$, where $\mathbf{P}_s = P_0\mathbf{n}$; (iii) P_0 is constant; (iv) dielectric tensor is isotropic, $\epsilon = \epsilon\mathbf{I}$; and (v) the orientation of the director at the surface is the same as prescribed by photopatterning.

In the formula for \mathbf{P} there are terms associated with the electric field induced polarization and spontaneous polarization P_s , however the LC director deformation may also result in the polarization. Why this formula does not include the flexopolarization?

By assuming P_0 is constant, we neglected the part of changes in the polarization value due to the flexoelectric effect. The values of spontaneous polarization in the ferroelectric phase are large, so it is expected that the small change in its value due to the flexoelectricity can be neglected in our simplified approach. This assumption is also supported by the results of the SHG microscopy, where in the N_F we did not notice significant changes in SHG intensity in the parts of the sample with more splay.

To include the changes in the polarization value due to the flexoelectric effect, the $P_0(\mathbf{r})$ should be obtained together with $\mathbf{n}(\mathbf{r})$ by the minimization of the free energy in which also other polarization terms are included (eg, $P_0^2, P_0^4, (\nabla P_0)^2$) similarly, as it was done in Refs (Mertelj, A. et al. Phys. Rev. X 8, 041025 (2018) & Sebastián, N. et al. Phys. Rev. Lett. 124, 037801 (2020)). Then the changes in the value of $P_0(\mathbf{r})$ due to flexopolarization come as a consequence of the flexoelectric term $-\gamma P_0(\mathbf{r}) \nabla \cdot \mathbf{n}(\mathbf{r}) \equiv -K_1 \mathbf{S}(\mathbf{r}) \cdot \mathbf{S}_0(\mathbf{r})$.

To clarify this point also for the readership, we included the following additional text in the main manuscript:

By assuming P_0 is constant, we neglected the part of changes in the polarization value due to the flexoelectric effect. The values of spontaneous polarization in the ferroelectric phase are large, so it is expected that the small change in its value due to the flexoelectricity can be neglected in our simplified approach. This assumption is also supported by the results of the SHG microscopy, where in the N_F we did not notice significant changes in SHG intensity in the parts of the sample with more splay.

2. I think that the reader would benefit if the authors explicitly write down the Poisson-Boltzmann equation here.

We agree with the referee and the Poisson-Boltzmann equation has been included in the main manuscript in the following way:

Assuming that the dynamics of \mathbf{n} (and \mathbf{P}_s) are much slower than that of free charges, then, during \mathbf{n} relaxation, the local field \mathbf{E} is given by the Poisson-Boltzmann equation $\nabla^2 \Phi_n = \beta^2 \sinh \Phi_n - \rho_{b,n}$ where $\Phi_n = e\phi/k_B T$, being $\phi(\mathbf{r})$ the electrostatic potential, and β and $\rho_{b,n}$ account for the normalized free charges and volume charges respectively.

And

The relaxation steps were performed with respect to $\vartheta(x, z)$ and $\varphi(x, z)$. At each step, \mathbf{E} was recalculated using the linearized Poisson-Boltzmann equation (for $\Phi_n < 1$, $\nabla^2 \Phi_n = \beta^2 \Phi_n - \rho_{b,n}$).

3. The free energy written below seems missing the terms describing the flexopolarization and induced polarization interaction with electric field. Let us suppose that the spontaneous polarization is zero ($P_0=0$), than from the equation below it is seen that there is no interaction with the electric field at all. While there should the electric field interaction with the induced polarization and with the flexopolarization.

to minimize the relevant part of the Landau-de Gennes free energy:

$$F_{\mathbf{n}} = \int \left(\frac{1}{2} K_1 |\mathbf{S} - \mathbf{S}_0|^2 + \frac{1}{2} K_2 T w^2 + \frac{1}{2} K_3 |\mathbf{B}|^2 - \frac{1}{2} P_0 \mathbf{n} \cdot \mathbf{E} \right) dV$$

Here, K_i ($i = 1, 2, 3$) are splay, twist, and bend elastic constants with corresponding deformations $\mathbf{S} = \mathbf{n} \nabla \cdot \mathbf{n}$, $T w = \mathbf{n} \cdot (\nabla \times \mathbf{n})$, $\mathbf{B} = \mathbf{n} \times (\nabla \times \mathbf{n})$, and $\mathbf{E} = -\nabla \Phi$. The flexoelectric coupling is added to the first term, where $\mathbf{S}_0 = \gamma \mathbf{P} / K_1$. The sign of the flexoelectric coefficient γ determines which direction of \mathbf{P} is favourable when a splay deformation \mathbf{S} is present.

In the simple model, in the written part of the free energy only the terms that depend on the orientation of \mathbf{n} are included, ie **the relevant** part of the free energy. As explained in the response to comment 1), the flexoelectric part of the polarization was neglected by assuming P_0 to be constant. Due to the assumption that the dielectric constant is isotropic, the induced polarization is parallel to the local field (resulting in a

term independent from \mathbf{n} : $-\frac{1}{2}\varepsilon\varepsilon_0\mathbf{E}^2$), which means that the torque of the local field on the induced polarization is 0. If the dielectric tensor is taken anisotropic, i.e. $\boldsymbol{\varepsilon} = \varepsilon_{\perp}\mathbf{I} + \Delta\varepsilon(\mathbf{n} \otimes \mathbf{n})$, there would be an additional term $-\frac{1}{2}\Delta\varepsilon\varepsilon_0(\mathbf{n} \cdot \mathbf{E})^2$, which is orders of magnitude smaller than the term $-\frac{1}{2}P_0\mathbf{n} \cdot \mathbf{E}$, due to the large polarization values ($\frac{P_0}{\varepsilon_0} \sim 5 \cdot 10^9$ V/m). So from this point of view, neglecting the dielectric anisotropy is justified.

If $P_0 = 0$, the expression simplifies to the situation of an apolar nematic in which flexoelectric coupling is neglected. To account for it, $P_0(\mathbf{r})$ should be treated as a variable in the minimization as explained in response to comment 1). To get the interaction with the field, the anisotropy of the dielectric tensor as described above needs to be taken into account.

To clarify this point also for the readership, description of the relevant part of the free energy now reads as:

Then the relaxation method can be used to minimize the part of the Landau-de Gennes free energy that depends on the director orientation:

4. It looks like that in the manuscript (see the equation for $F_{\mathbf{n}}$ shown above) and Supplementary Information (eqn. (11)) S_0 is a vector, on the other hand the Supplementary Information says $S_0 = \gamma \mathbf{P} \cdot \mathbf{n}$, which means S_0 is a scalar.

$$F_{\mathbf{n}} = \int \left(\frac{1}{2}K_1|\mathbf{S} - \mathbf{S}_0|^2 + \frac{1}{2}K_2Tw^2 + \frac{1}{2}K_3|\mathbf{B}|^2 - \frac{1}{2}P_0\mathbf{n} \cdot \mathbf{E} \right) dV. \quad (11)$$

The relaxation steps were performed with respect to $\vartheta(x, z)$ and $\varphi(x, z)$. At each step, \mathbf{E} was recalculated using Eq.7. Here, K_i ($i = 1, 2, 3$) splay, twist, and bend elastic constants with corresponding deformations $\mathbf{S} = \nabla \cdot \mathbf{n}$, $Tw = \mathbf{n} \cdot (\nabla \times \mathbf{n})$, $\mathbf{B} = \mathbf{n} \times (\nabla \times \mathbf{n})$, and $\mathbf{E} = -\nabla\phi$. The flexoelectric term is included in the first term, where $S_0 = \gamma \mathbf{P} \cdot \mathbf{n} / K_1$ is the ideal splay curvature, which would minimize the splay elastic energy.

We agree with the referee that this can induce to confusion. Because of the assumption $\mathbf{P}_s = P_0\mathbf{n}$ both expressions are equivalent, however, for consistency, we change SI to mirror expression in the main manuscript.

5. To calculate the electric field the authors assume that the LC is isotropic fluid, the LC dielectric tensor is isotropic and has only one component. In fact, the LC possesses the dielectric anisotropy, $\boldsymbol{\varepsilon}$ is a tensor. The LC dielectric anisotropy can be rather high. This anisotropy may significantly affect the local electric field, both the magnitude and the spatial profile of $\mathbf{E}(\mathbf{r})$. Isn't the above assumption an oversimplification? How big is the expected error when the LC is assumed to be an isotropic medium?

As mentioned above, $\mathbf{P} = \mathbf{P}_s + \varepsilon_0(\boldsymbol{\varepsilon} - \mathbf{I})$ and the dielectric tensor is taken anisotropic, i.e. $\boldsymbol{\varepsilon} = \varepsilon_{\perp}\mathbf{I} + \Delta\varepsilon(\mathbf{n} \otimes \mathbf{n})$. That is, the induced polarization anisotropy is neglected.

Dielectric properties in ferroelectric nematic materials is certainly a challenging topic and poorly understood at the moment. There are limited experimental investigations and to our knowledge, no values of dielectric anisotropy have been yet reported. This is understandable due to the difficulties to perform systematic investigations. The induced contribution is smaller than the spontaneous polarization and in experiments, both contributions are present and separating them is challenging.

The simplified model presented in this manuscript qualitatively describes the system, and is explored to assess the stability of the structures deduced from the microscopy investigations. Together with the lack of reliable dielectric anisotropy data at this moment, we opted for the most sensible approach of simplifying the model and consider it isotropic.

In order to estimate the error when adopting this assumption, we compared the electrostatic potentials calculated using the finite element method for $\varepsilon_{\perp} = 100$ and anisotropies $\Delta\varepsilon = 0, 10, \pm 50$, and 100 for the

structure given in Supplementary Note VII, (Expression (1)–(4)) using the general Poisson-Boltzmann equation $1/\varepsilon_{\perp} \nabla \cdot \boldsymbol{\varepsilon} \cdot \nabla \Phi_n = \beta^2 \sinh \Phi_n - \rho_{b,n}$ or its linearized version $1/\varepsilon_{\perp} \nabla \cdot \boldsymbol{\varepsilon} \cdot \nabla \Phi_n = \beta^2 \Phi_n - \rho_{b,n}$. Because the criteria $\Phi_n < 1$ is met, both gave almost identical results. The difference between the calculated normalized potentials for the anisotropic and isotropic case ($\Delta\varepsilon = 0$) is as expected most pronounced where the angle ϑ is the largest. However, even there, the difference is small, i.e. below 1.3%, 6.2%, and 13% for $\Delta\varepsilon = 10, \pm 50,$ and 100 , respectively. This suggests that accounting for the anisotropy would not qualitatively change the results.

A remark about this has been added in the SI Note VII:

i) S is constant, so the nematic order can be described only by \mathbf{n} ; (ii) $\mathbf{P} = \mathbf{P}_s + \varepsilon_0(\boldsymbol{\varepsilon} - \mathbf{I})\mathbf{E}$, where $\mathbf{P}_s = P_0\mathbf{n}$; (iii) P_0 is constant; (iv) induced polarization anisotropy is neglected and dielectric tensor is thus taken as isotropic, $\boldsymbol{\varepsilon} = \varepsilon\mathbf{I}$; and (v).....

Additionally we added the following clarification into the SI:

If the dielectric tensor is taken anisotropic, i.e. $\boldsymbol{\varepsilon} = \varepsilon_{\perp}\mathbf{I} + \Delta\varepsilon(\mathbf{n} \otimes \mathbf{n})$, there would be an additional term $-\frac{1}{2}\Delta\varepsilon\varepsilon_0(\mathbf{n} \cdot \mathbf{E})^2$, which is orders of magnitude smaller than the term $-\frac{1}{2}P_0\mathbf{n} \cdot \mathbf{E}$, due to the large polarization values ($\frac{P_0}{\varepsilon_0} \sim 5 \cdot 10^9$ V/m). So from this point of view, neglecting the dielectric anisotropy is justified.

6. To calculate the electric potential the authors linearize the Poisson-Boltzmann equation.

the electrostatic potential can be calculated using Poisson-Boltzmann equation

$$\nabla^2 \Phi_n = \beta^2 \sinh \Phi_n - \rho_{b,n} \quad (6)$$

Here, $\Phi_n = e\phi/k_B T$. If $\Phi_n < 1$, as an approximation, a linearized Poisson-Boltzmann equation

$$\nabla^2 \Phi_n = \beta^2 \Phi_n - \rho_{b,n} \quad (7)$$

How do we know that? An estimate would be useful here.

Linearized Poisson-Boltzmann (PB) equation is valid provided that the screening length $\lambda_D = \sqrt{(\varepsilon\varepsilon_0 k_B T)/(2n_0 e^2)}$ is smaller than the features of the system. For the studied systems the smallest features are located at the edge of the splay ($x = \pm L/2$) with a length scale of 100 nm, so the only case in which the use of linearized Poisson-Boltzmann equation is questionable is for $\varepsilon = 1000$ and $\beta^2 = 1$ for which $\lambda_D = 200$ nm. In this case, Φ_n exceeds 1 only at the edges of the splay, where it reaches the value of 1.5. Comparison of the solution of Poisson-Boltzmann equation with linearized equation shows that this accounts for an error of around 10% in that region.

We added the following text in Supplementary Information:

The linearized Poisson-Boltzmann equation is valid provided that the screening length λ_D is smaller than the features of the system and ξ_b .

This point has been clarified in the Supplementary Fig.20 caption, under the relevant calculations.

For $\varepsilon = 1000$ and $\beta^2 = 1$, $\lambda_D = 200$ nm and Φ_n exceeds 1 only at the edges of the splay $\sim \pm L/2$, where it reaches the value of 1.5. Comparison of the solution of general Poisson-Boltzmann equation with linearized equation, shows that this accounts for an error of around 10% in that region.

7. Minor comment. "... described by R.B. Meyer,¹⁸ can be illustrated considering a nematic phase formed

by pearshaped or bent-shaped. “ Originally Meyer (ref [18]) called them wedge and crescent shape, see the picture from his work [18].

FIG. 1. (a) and (c) Unstrained nematic structures containing polar molecules. (b) Splayed structure in which splay and polarization are coupled by the wedge shape of the molecule. (d) Bent structure in which the bending and polarization are coupled by the crescent shape of the molecule.

The text has been modified accordingly and now reads as:

“originally described by R.B. Meyer [18] can be illustrated considering a nematic phase formed by wedge-shaped or crescent-shaped (bent-shaped) molecules with dipole moment along the long and short axis, respectively”

REVIEWERS' COMMENTS

Reviewer #2 (Remarks to the Author):

The authors have revised the manuscript and now it can be accepted for publication.

Reviewer #3 (Remarks to the Author):

I'm happy with the revisions made by the authors.